# Step-to-step variability indicates disruption to balance control when linking the arms and legs during treadmill walking

**Daisey Vega[1]☯, Helen J. Huang[2,3]☯, Christopher J. Arellano[1]☯ ***

**1** Department of Health and Human Performance, Center for Neuromotor and Biomechanics Research Laboratory, University of Houston, Houston, Texas, United States of America, **2** Department of Mechanical and Aerospace Engineering, University of Central Florida, Orlando, Florida, United States of America, **3** Disability, Aging, and Technology (DAT) Cluster, University of Central Florida, Orlando, Florida, United States of America

☯ These authors contributed equally to this work.
* carellano@uh.edu

**Data Availability Statement:** All relevant data are within the Supporting Information files.

**Funding:** The author(s) received no specific funding for this work.

## Abstract

We recently discovered that a rope-pulley system that mechanically coupling the arms, legs and treadmill during walking can assist with forward propulsion in healthy subjects, leading to significant reductions in metabolic cost. However, walking balance may have been compromised, which could hinder the potential use of this device for gait rehabilitation. We performed a secondary analysis by quantifying average step width, step length, and step time, and used their variability to reflect simple metrics of walking balance ($n = 8$). We predicted an increased variability in at least one of these metrics when using the device, which would indicate disruptions to walking balance. When walking with the device, subjects increased their average step width ($p < 0.05$), but variability in step width and step length remained similar ($p's > 0.05$). However, the effect size for step length variability when compared to that of mechanical perturbation experiments suggest a minimal to moderate disruption in balance (Rosenthal ES = 0.385). The most notable decrement in walking balance was an increase in step time variability ($p < 0.05$; Cohen's $d = 1.286$). Its effect size reveals a moderate disruption when compared to the effect sizes observed in those with balance deficits (effect sizes ranged between 0.486 to 1.509). Overall, we conclude that healthy subjects experienced minimal to moderate disruptions in walking balance when using with this device. These data indicate that in future clinical experiments, it will be important to not only consider the mechanical and metabolic effects of using such a device but also its potential to disrupt walking balance, which may be exacerbated in patients with poor balance control.

## Introduction

Arm movements are beneficial for walking and can play an integral part in gait rehabilitation [1–4]. These insights motivated us to develop a simple rope-pulley system in which the active use of the arms is mechanically linked to the legs during treadmill walking, with the hope that

**Competing interests:** The authors have declared that no competing interests exist.

this system could serve as a gait rehabilitation strategy. We discovered that the active use of the arms with this simple device helped propel the body by assisting with the generation of propulsive forces, which led to a significant reduction in the net metabolic power required to walk [5]. Reducing net metabolic power has important clinical implications because individuals with gait pathologies may experience major decrements in walking economy [6], signifying that it is highly strenuous to walk. While there are various factors that may impair walking economy, one critical factor is the inability to generate propulsive forces by the legs, which has been observed in individuals recovering from a spinal cord injury [6]. Therefore, the active use of the arms by means of our rope-pulley system may help alleviate this mechanical demand. By assisting in the generation of propulsive forces, the active use of the arms can make it easier to propel the body in the forward direction.

Despite the mechanical and metabolic benefits and their potential clinical implications, we also observed abnormal arm swing with the use of our device. Abnormal arm swing was characterized by increased shoulder protraction and increased elbow joint flexion/extension. This might be an undesirable trade-off since a recent experiment found that abnormal increases in arm swing, with amplitudes reaching shoulder height, increases stepping variability [7]. The increase in stepping variability suggests a possible disruption to walking balance. While our simple rope-pulley device provides both mechanical and metabolic benefits [5], this may have come at the expense of increased stepping variability, which would indicate an undesirable disruption to walking balance. It is well documented that humans maintain walking balance by adjusting their foot placement and timing from step to step [6, 8, 9]. The variability in one's stepping pattern can serve as simple metrics of walking balance [10–12], which have good validity for predicting fall probability [12] and are good clinical correlates for walking balance [13]. For example, the variability in step width, step length and/or step time have been used as simple metrics of balance in patients recovering from a spinal cord injury [13], patients diagnosed with Parkinson's disease [14] and peripheral neuropathy [15, 16], and older adults experiencing multiple falls [17]. Therefore, to understand if mechanically linking the active use of the arms with the legs disrupted walking balance, we carried out a secondary analysis by quantifying the variability in step width, step length, and step time.

Understanding if, and to what extent, balance was compromised will help determine if linking the arms and legs during treadmill walking could feasibly translate to clinical users undergoing gait rehabilitation. Helping patients maintain balance while practicing their stepping motion is critical for walking recovery [18, 19], reducing fear of falling [20, 21], and preventing falls [22]. Ideally, our arm-leg rope pulley system would not disrupt walking balance, but our observations of abnormal arm swinging suggest otherwise. Therefore, we wanted to understand if physically coupling of the arms and legs during walking may have influenced balance. Given the novelty of this device, we did not have any *a priori* knowledge as to which stepping parameter(s) would exhibit an increase in variability. For this reason, we hypothesized that walking with our arm-leg rope pulley system would increase the variability in either step width, step length, and/or step time. If walking with our arm-leg rope pulley system did disrupt balance, we expect an increase in the variability of at least one of these stepping parameters. We also suspected that subjects could adjust their foot placement strategy when walking with the rope pulley system, which may or may not be coupled with an increase in variability. Therefore, we examined whether subjects altered their average step width, step length, and/or step time while walking with the arm-leg rope pulley system, as this could indicate a foot placement strategy to avoid disruptions to walking balance.

## Methods

Eight healthy subjects (3 females and 5 males; mean ± SD: age = 23.25 ± 3.37 years, mass = 73.88 ± 18.46 kg, height = 173.84 ± 13.95 cm) provided consent and participated in the

study, which was approved by the University of Houston Institutional Review Board. Subjects walked on a dual-belt treadmill at 1.25 m/s for randomized conditions of walking with the arm-leg rope-pulley system (assisted walking; Fig 1) and without it (normal walking). Each trial lasted 7 minutes with at least 5 minutes of rest between trials.

From our previous data [5], we calculated step width, step length, and step time from the positions of the left and right heel markers. We used the vertical versus time component of each heel marker to identify initial contact as instances when the waveform exhibited a trough during each step. At each instance of initial contact, we quantified step width and step length as the medio-lateral (ML) and anterior-posterior (AP) distance between the left and right heel markers, respectively, and step time as the period between consecutive instances of initial contact. We determined the number of steps achieved by each subject and then found that 288 steps were the minimum number of steps taken across all subjects. For every trial in our analysis, we used these 288 steps to calculate the average and standard deviation values for all step parameters. For each subject, we normalized step width, step length, and their variability by dividing each metric by leg length (LL) and expressed these values as a percentage [8, 23]. Leg length is defined as the distance between the greater trochanter and the ground. We also normalized step time by $\sqrt{LL/g}$ where g is gravity [23]. To help visualize the time series data for each stepping parameter across 288 steps, Fig 2 illustrates the time series data for one subject.

For our statistical analysis, the Shapiro-Wilk was used to test the assumption of normality. If the assumption of normality was met, we performed paired sample $t$-tests to test for significant differences between normal and assisted walking conditions. The values for each condition are reported as mean ± standard deviation (SD). Additionally, Cohen's $d$ was used to calculate a parametric effect size (noted as Cohen ES). If the assumption of normality was not met, we used the non-parametric Related Samples Wilcoxon Signed Rank test. In this case, these values are reported as median values with their interquartile range (IQR), defined as the range between the 25th and 75th percentile. Furthermore, a non-parametric effect size noted as Rosenthal ES [24] was calculated by diving the Z score by the square root of N where N is the number of observations for both conditions (i.e. N = 16). In line with our hypothesis, we used a one-tailed test for the variability metrics and a two-tailed test for the average metrics. All statistical tests were performed with an $\alpha$-level of 0.05 (IBM SPSS Inc., Chicago, IL, USA).

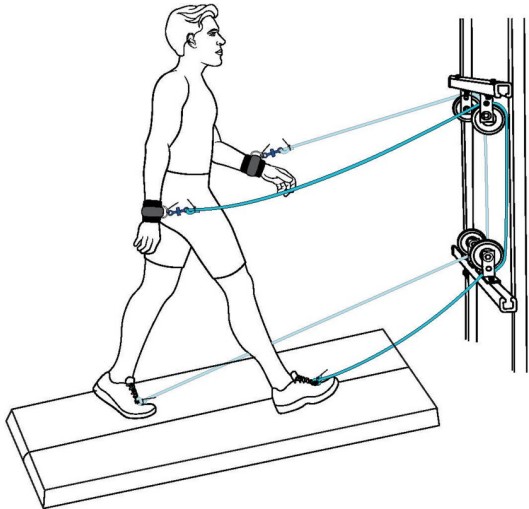

**Fig 1. Arm-leg rope pulley system.** Subjects walked on a treadmill while attached to a simple rope-pulley device that connects the ipsilateral arm and leg via a rope. Figure modified from Vega and Arellano [5].

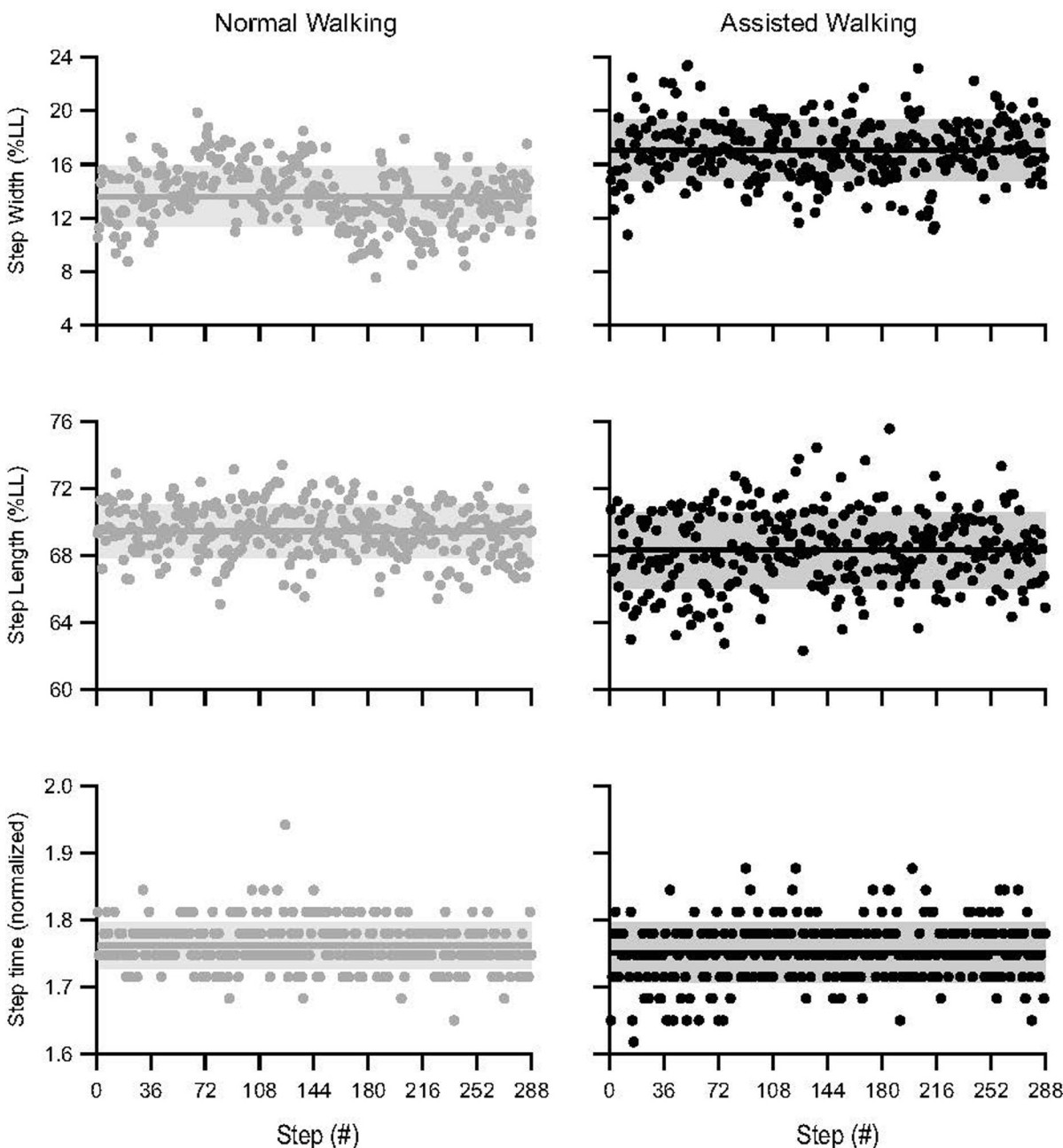

**Fig 2. Time series data (*n* = 1).** Solid circles represent values for each step over a series of 288 consecutive steps for one subject. Average (solid line) ± 1 SD (shaded region) are shown for each stepping parameter during normal and assisted walking conditions. When compared to normal walking, the subject increased their average step width by 0.032 m (normalized: 3.45% LL) during assisted walking, but step width variability remained relatively unchanged (2.27% LL vs 2.26% LL). Additionally, this subject slightly decreased their average step length by 0.011 m (normalized: 1.13% LL) and exhibited a 0.006 m (normalized: 0.69% LL) increase in step length variability. Lastly, the subject did not alter their average step time (54 ms vs 54 ms), but their step time variability slightly increased by 3.2 ms (normalized: 0.01). Note that all subjects responded in a slightly different way (see Fig 3 for individual data).

## Results

All variables met the assumption of normality except for step length variability and average step width. Average step length (two-tailed *p* = 0.718, Cohen ES = 0.133) and step time (two-

tailed $p$ = 0.233, Cohen ES = 0.461) revealed no significant differences between normal and assisted walking conditions. Yet, when compared to normal walking, subjects increased their average step width by 0.032 m during assisted walking (normalized median ± IQR: 14.42 ± 5.67% LL vs 18.83 ± 5.69% LL; Wilcoxon Signed Rank test, two-tailed $p$ = 0.012, Z = 2.521, Rosenthal ES = 0.630, Fig 3). Despite adjustments in step width, there were no statistical differences in either step width variability (one-tailed $p$ = 0.347, Cohen ES = 0.146) or step length variability (Wilcoxon Signed Rank test, one-tailed $p$ = 0.062, Z = 1.540, Rosenthal ES = 0.385). In contrast, walking with the assistive arm-leg device increased step time variability by an average of 5 ms (normalized mean ± SD: 0.04 ± 0.01 vs. 0.06 ± 0.01; one-tailed $p$ = 0.004, Cohen ES = 1.286; Fig 3). For reference, the data underlying the findings of this study are made available in the S1 File.

## Discussion

From the balance metrics quantified here, we found that linking the active use of the arms with the legs during treadmill walking led to a significant increase in step time variability, but not step width nor step length variability. These findings support our hypothesis that walking with our arm-leg pulley system would increase the variability of at least one of these stepping parameters. To assess whether the statistical increase in step time variability reflected a meaningful change, we used its effect size to interpret the relative magnitude to which walking balance was disrupted. Additionally, the value of calculating effect sizes is that they are independent of sample size and allows us to compare findings from various studies. Our literature review revealed that data on step time variability in individuals recovering from a spinal cord injury or stroke were not available and therefore, we could not make effect size comparisons to the intended population who could possibly benefit from such a device. Instead, we gathered data on individuals with balance deficits manifesting from neurological disorders and aging as well as data from mechanical perturbation experiments (S1 File).

In individuals with Parkinson's disease and older adults experiencing multiple falls, step time variability is greater than healthy control subjects (Hedges' g = 1.509 [14]) and non-fallers (Cohen ES = 0.486 [17]), respectively. For relative comparisons, these effect sizes ranged between 0.486 and 1.509, and our observed effect size for step time variability (Cohen ES = 1.286) falls within this range. While these comparisons are to be taken with caution, they are still useful for gauging the degree to which our rope-pulley system disrupts walking balance. Therefore, we interpret our effect size for step time variability to indicate a moderate disruption to the temporal component of walking balance.

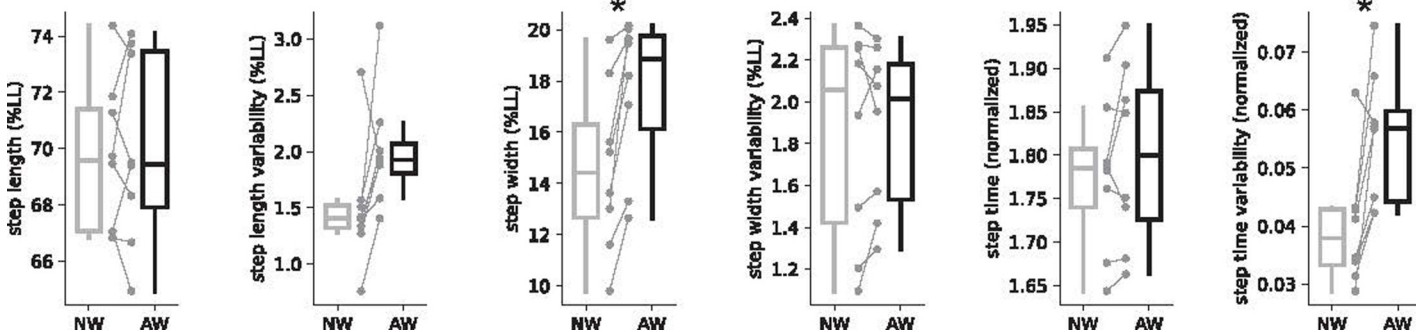

**Fig 3. Box and whisker plots for all stepping parameters during normal (NW) and assisted (AW) walking conditions.** Individual data points are superimposed and connected to show paired data ($n$ = 8). Most notably, all subjects increased their average step width. Additionally, seven out of eight subjects exhibited an increase in step time variability. Asterisks indicate $p < 0.05$.

As for the spatial component of walking balance, we did not detect a statistically significant increase in either step width variability or step length variability. Given our small sample size, it is helpful to examine more closely the individual trends in stepping variability. For instance, half of our subjects exhibited a small increase in step width variability while the other half exhibited a small decrease (Fig 3). Overall, the individual changes in step width variability appear small, but for relative comparisons, we can judge our findings to that of Madehkhaksar et al. [25], who applied sudden mechanical perturbations in the forward direction and observed significant increases in step width variability during treadmill walking in healthy subjects. Based on our estimate from their published data, the increase in step width variability reflects a Cohen ES of 0.845. Our observed Cohen ES for step width variability is 0.146, falling well below the value that is observed from an experiment that intentionally induced mechanical perturbations in the forward direction. Collectively, our results suggest that the arm-leg rope pulley system elicited little to no disruption to balance in the medio-lateral direction. On the contrary, seven out of eight subjects exhibited an increase in step length variability (Fig 3). With limited step length variability data in the literature, we were only able to make an effect size comparison to the 12-month retrospective study of Callisaya et al. [17], who reported differences between older adults who experience multiple falls and to those with no falls (Hedges' g = 0.369). Our observed effect size for step length variability (Rosenthal ES = 0.385) is similar in magnitude. Therefore, given the relative magnitude of our effect size, irrespective of statistical non-significance, it is reasonable to consider the increase in step length variability as a meaningful disruption to balance. In all, we conclude that linking the active use of the arms with the legs via our simple rope-pulley system led to a minimal to moderate disruption to balance in the anterior-posterior direction.

These comparisons should be interpreted in the context of our small sample size and the limited data that is reported for stepping variability in the clinical and gait rehabilitation literature. Differences in normalization techniques, or lack thereof, further reduce the amount of available literature we could compare. In addition, it was not possible to account for differences in walking speed or conditions (e.g., overground versus treadmill walking), which influence these stepping parameters as well. It should also be noted that the analyses of this study were focused on linear analyses of gait variability. It may be helpful to consider other metrics (e.g., fractal analysis of gait variability, Lyapunov exponent, etc.) that are based on non-linear analyses in future studies, which may capture potential disruptions to balance in ways that linear analyses do not. Despite these limitations, we use effect size comparisons as a reasonable gauge to help interpret the degree which balance control may have been disrupted.

To further address the limitation of our small sample size, we carried out sensitivity analyses in GPower (please see S2 File for more details) to determine if the findings from our variability variables were adequately powered. Our analysis parameters were set to 80% power, α = 0.05 and a one-tailed dependent t-test or a one-tailed Wilcoxon signed rank test. Based on the analyses, a sample size of 8 indicates that the minimum effect size that can be detected are 0.978 for step width variability and 1.007 for step length variability. The effect sizes found in our study (step width variability ES = 0.146 and step length variability ES = 0.385) fall short of the minimum effect size that can be detected with 80% power. Therefore, we note that our findings for step width and length variability may not be adequately powered with our sample size of 8. However, the sensitivity analysis reveals that our findings for step time variability appears to be adequately powered since our effect size (1.286) is above the minimum effect size of 0.978 detectable at 80% power. We interpret this to suggest that the temporal component of walking balance is most disrupted with the use of the arm-leg rope-pulley system. Yet, it is possible that the spatial component may have been disrupted but was not detectable with our sample size of 8 subjects.

To determine the appropriate sample size needed for a future study, we carried out another sensitivity analysis (S2 File). For this analysis, we used "meaningful" effect sizes based on data gathered from the literature to determine the sample size needed for a study with sufficient power (parameters were set to 80% power, $\alpha = 0.05$, and a one-tailed dependent t-test). The meaningful effect size for each stepping variable were gathered from two studies that investigated stepping variability in fallers versus non-fallers [17] and under treadmill perturbations [25]. Given the limited data, these studies were chosen as a reasonable gauge because their findings suggest that 1) increases in step time and step length variability are most notable in distinguishing fallers from non-fallers [17] and 2) step width variability increased under a forward treadmill perturbation [25], which is analogous to the effects of our arm-pulley device. This sensitivity analysis reveals that the sample size for a sufficiently powered study would require between 10 to 49 subjects. We suspect that for individuals undergoing gait rehabilitation (e.g., patients with a spinal cord injury), such effect sizes would be much greater in magnitude, and therefore, this may be a conservative estimate of sample size. A sample size of 49 is atypical in experiments involving individuals with an incomplete spinal cord injury, but a sample size of 10 may be too small. Considering the resources, time, and challenges with subject recruitment of this clinical population [26], we feel that a sample size of $n = 20$ would be reasonable for a future study investigating walking balance using linear variability measures.

In addition to increasing the sample size for future experiments, we also aim to modify our arm-leg pulley system and experimental design so that it minimizes any disruption to the subject's preferred stepping kinematics. When considering the other foot placement metrics quantified here, we found that healthy subjects increased their average step width, which may reflect a foot placement strategy to maintain an appropriate level of balance in the medio-lateral direction. Given the novelty of using the device, it is possible that linking the arms and legs during treadmill walking may have induced a psychological fear of falling, which in itself has been associated with increases in step width [21]. It is also possible that the arm-leg rope pulley system was too wide for subjects, leading them to compensate by increasing their step width. The short and single-session familiarization period may have also contributed to subjects taking wider steps. Adopting wider steps could be a response to learning under a novel condition where maintaining balance is more challenging and therefore, requires an individual to alter their stepping strategy [27]. For future experiments, we will incorporate longer sessions that allow for learning adaptation and adjust the width of both pulleys so that they are aligned to the subject's step width. These modifications may help to mitigate disruptions to walking balance and thus, minimize the need for subjects to alter their foot placement strategy in the medio-lateral direction.

In conclusion, when physically linking the arms and legs during treadmill walking, healthy subjects exhibited the most notable increase in step time variability, and its effect size reveals a moderate disruption to the temporal component of balance. There also appears to be a minimal to moderate disruption to the spatial component of walking balance, particularly along the anterior-posterior direction. While disruption to walking balance ranges between minimal to moderate in healthy subjects, the question remains as to whether using this device would exacerbate disruptions to walking balance in individuals with already poor balance control, such as those recovering from a spinal cord injury or stroke. Negating disruptions to walking balance would allow these patients to focus on coordinating the active use of their arms with the stepping motion of their legs during gait retraining. Maintaining an appropriate level of walking balance while benefiting from the mechanical and metabolic effects will be key to understanding whether this device or a modified version can promote walking recovery in a clinical setting.

## Supporting information

**S1 File. Data set and effect sizes.** This excel file contains the data set for this study as well as the calculated effects sizes from literature reported in the discussion.
(XLSX)

**S2 File. Sensitivity analyses.** This file contains the protocol and graphs of the sensitivity analyses performed in GPower.
(DOCX)

## Acknowledgments

We would like to thank Dr. Stacey L. Gorniak and Dr. Adam Thrasher for their helpful feedback on previous versions of this manuscript. We are also grateful to Dr. Daniel P. O'Connor for his expert advice on statistical power and sensitivity analyses.

## Author Contributions

**Conceptualization:** Daisey Vega, Helen J. Huang, Christopher J. Arellano.

**Data curation:** Daisey Vega, Helen J. Huang, Christopher J. Arellano.

**Formal analysis:** Daisey Vega, Helen J. Huang, Christopher J. Arellano.

**Funding acquisition:** Christopher J. Arellano.

**Investigation:** Daisey Vega, Christopher J. Arellano.

**Methodology:** Daisey Vega, Helen J. Huang, Christopher J. Arellano.

**Project administration:** Christopher J. Arellano.

**Resources:** Christopher J. Arellano.

**Software:** Daisey Vega, Helen J. Huang, Christopher J. Arellano.

**Supervision:** Christopher J. Arellano.

**Validation:** Christopher J. Arellano.

**Visualization:** Daisey Vega, Helen J. Huang, Christopher J. Arellano.

**Writing – original draft:** Daisey Vega, Christopher J. Arellano.

**Writing – review & editing:** Daisey Vega, Helen J. Huang, Christopher J. Arellano.

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
