## [Decision Letter · Decision Letter 0]

1 Nov 2021

PONE-D-21-31057Step-to-step variability indicates minimal disruption to balance when linking the arms and legs during treadmill walkingPLOS ONE

Dear Dr. Arellano,

Thank you for submitting your manuscript to PLOS ONE. After careful consideration, we feel that it has merit but does not fully meet PLOS ONE’s publication criteria as it currently stands. Therefore, we invite you to submit a revised version of the manuscript that addresses the points raised during the review process.

We look forward to receiving your revised manuscript.

Kind regards,

David J Clark

Academic Editor

PLOS ONE

a) Did participants provide their written or verbal informed consent to participate in this study?

Reviewers' comments:

Reviewer's Responses to Questions

**Comments to the Author**

1. Is the manuscript technically sound, and do the data support the conclusions?

Reviewer #1: Yes

Reviewer #2: No

2. Has the statistical analysis been performed appropriately and rigorously? 

Reviewer #1: Yes

Reviewer #2: Yes

3. Have the authors made all data underlying the findings in their manuscript fully available?

Reviewer #1: Yes

Reviewer #2: Yes

4. Is the manuscript presented in an intelligible fashion and written in standard English?

Reviewer #1: Yes

Reviewer #2: Yes

5. Review Comments to the Author

Reviewer #1: In this paper the authors conduct a secondary analysis on a device they previously developed that may be useful as a gait rehabilitation device in the future. Specifically, they previously saw metabolic benefits to mechanically coupling the arms and the legs during treadmill walking, but this new device also lead to abnormal arm swing. Thus, in the present study the authors sought to determine if there were changes in the stepping pattern while mechanically coupling the arms and legs since significant disruptions to normal walking could limit the device’s usefulness in a patient population.

Overall, I am satisfied with the manuscript as it was initially submitted, and I have no major suggestions for improvement. My recommendation is to accept the paper. The authors frame the scope of this paper well. The methods are sound, and the conclusions they draw are reasonable. I look forward to seeing the future work. My three comments below are minor and do not need to be addressed in the paper itself. They are more curiosities than critiques.

Comment 1: I’m curious what the effect size was for step length variability. The p-value was not quite significant, but Figure 3 shows a decent effect in all but one subject, as you mentioned in the caption. In the ideal world, your device would provide metabolic benefits without any changes to the stepping pattern. Thus, for the purposes of this paper (identifying potential disruptions to normal walking), I think it is reasonable to consider a p-value of 0.062 as a potential effect. Step length and step length variability are the variables I would a priori assume to be most affected (as opposed to step width) since the rope-pulley system directly pulls on the feet in the anteroposterior direction as opposed to mediolaterally.

Comment 2: In future work, I would be interested in exploring connecting the arms to the contralateral legs. You referenced previous work by Zehr who has explored the neural coupling between the arms and legs. I am only somewhat aware of his work, not the major findings nor how they directly relate to your current work. I would be interested to know from a neuro-rehab perspective if linking arms and legs ipsilaterally versus contralaterally is expected to be more beneficial, for example for individuals that have experienced a stroke. It would be slightly more difficult to implement than your current setup, but it should still be reasonably possible to change the mechanical coupling to be contralateral if there were a specific benefit to connecting them that way.

Comment 3: In future work I might also suggest adding an elastic band or other spring mechanism between the ropes and the contact points on the body. That would allow you to dampen or limit the forces between the arms and legs which may help people more quickly learn how to walk in the device.

Reviewer #2: • Briefly describe the rationale and mechanism of the rope-pulley system. It seems to be beneficial for reducing the mechanical cost of walking based on a previous study, but there is no description of why it is important and by what mechanism it helps human gait. Readers should understand this from the current paper without visiting the previous study. Briefly describe this in the introduction.

• Other questions regarding the rope-pulley system: Why is reducing net metabolic power important when walking? Does the system increase arm swing range of motion or velocity?

• I have a concern regarding the statistical power of this study. Although the difference in stride length variability was insignificant, it was close to be significant, and all but one participant showed increased stride length variability (Figure 3). I suspect a low statistical power of this result along with low effect size in other results and thus suggest increasing the sample size or justify the sample size (n = 8) by conducting a power analysis.

• The summary of findings based on the low sample size also raises concerns. The authors repeatedly states that there was no increase in either step width variability or step length variability which indicate no spatial components of walking balance. This conclusion is a big leap and could be biased without warranting the statistical power.

• The authors’ interpretation of small effect size also raises concerns. The authors interpret the small effect size for step time variability to indicate a minimal disruption to the temporal component of walking balance. However, the small effect size could be due to the small sample size not because of the minimal disruption. Verify that the degree of differences in step time variability is minimal before reaching to a conclusion.

6. PLOS authors have the option to publish the peer review history of their article (what does this mean?). If published, this will include your full peer review and any attached files.

Reviewer #1: No

Reviewer #2: **Yes: **Kyoung Shin Park

---

## [Author Response · Author response to Decision Letter 0]

22 Nov 2021

For your reference, we have uploaded a "Response to Reviewers" document.

---

## [Decision Letter · Decision Letter 1]

19 Dec 2021

PONE-D-21-31057R1Step-to-step variability indicates disruption to balance control when linking the arms and legs during treadmill walkingPLOS ONE

Dear Dr. Arellano,

Thank you for submitting your manuscript to PLOS ONE. After careful consideration, we feel that it has merit but does not fully meet PLOS ONE’s publication criteria as it currently stands. Therefore, we invite you to submit a revised version of the manuscript that addresses the points raised during the review process.

One of the reviewers is suggesting additional data analysis.  I reached out to this reviewer for additional clarification/justification on the analysis approach that he proposed. The reviewer responded with an updated version of his comments, which I am pasting here in their entirety. Please consider adding to your analysis of the data, or justifying why additional analysis may not be warranted. The authors conducted a secondary data analysis from a parent study to further examine whether the rope pulley system disrupts walking balance in healthy adults. They hypothesized that the rope pulley system may increase the variability of at least one of step width, step length, and step time which may indicate the disruption of walking balance. The result indicates increased variability in step time (M = 5 ms, p < 0.05, Cohen’s d = 1.286) but not in other parameters. The authors conclude that healthy subjects experienced minimal to moderate disruptions in walking balance when using with this device. This study addresses an interesting topic, and the manuscript is well written. However, given that the analysis is based on a small sample size and the significant result was found in only one of three variables of interest, there remains a couple of shortcomings in this study to be published under the current condition.The finding reported in this study is that the rope pulley system marginally increased step time variability but not all other parameters of interest. I think this is comparably weak findings with a small sample size. The variability parameters are all linear variability and non-linear gait dynamics will be a great addition to the current weak findings.Please provide the rationale for the hypothesis. How can an increase in either one of three parameters of variability be enough to indicate disruption in walking balance?I understand that it would be not feasible to collect more data as this is a secondary data analysis. Anyhow, please report the power of current analysis which may provide more clear view of your acknowledgement of the small sample size as the limitation of this study.How were the variability parameters computed? Provide citation for the way of normalized standard deviation with leg length. Standard deviation also needs to be normalized with average value, which is a common computation of gait variability called coefficient of variation. Please consider using this parameter and justify the current data processing method. Please provide the interpretation of Figure 2. What do you mean by a representative subject?Standard deviation-based variability parameters are linear variability of human gait.  Please consider additional analysis of non-linear analysis of gait dynamics to strengthen the current findings. Non-linear gait analysis is appropriate for this type of time-series gait data. While the linear variability of gait represents the magnitude of the step-to-step fluctuation, dynamics of gait changes (alterations in the fractal pattern) is a great addition to the study of gait variability (Hausdorff, 2005, DOI: https://doi.org/10.1186/1743-0003-2-19). Although fractal analysis of gait variability used to conducted with time series gait data with > 600 strides, Kuznetsov and Rhea (2017) demonstrated that it is possible with relatively short gait time series (100 - 200 strides) (DOI: 10.1371/journal.pone.0174144). If this additional analysis of gait variability is not conducted, please provide justification that three variables of interest are adequate indicators of gait instability. Also discuss and acknowledge that the current analysis of gait variability is limited to linear analysis and that fractal analysis of gait variability was not considered due to some limitation but will strengthen the whole picture of gait variability altered by the rope pulley system in a future study.==============================

We look forward to receiving your revised manuscript.

Kind regards,

David J Clark

Academic Editor

PLOS ONE

Reviewers' comments:

Reviewer's Responses to Questions

**Comments to the Author**

1. If the authors have adequately addressed your comments raised in a previous round of review and you feel that this manuscript is now acceptable for publication, you may indicate that here to bypass the “Comments to the Author” section, enter your conflict of interest statement in the “Confidential to Editor” section, and submit your "Accept" recommendation.

Reviewer #1: All comments have been addressed

Reviewer #2: (No Response)

2. Is the manuscript technically sound, and do the data support the conclusions?

Reviewer #1: Yes

Reviewer #2: Partly

3. Has the statistical analysis been performed appropriately and rigorously? 

Reviewer #1: Yes

Reviewer #2: Yes

4. Have the authors made all data underlying the findings in their manuscript fully available?

Reviewer #1: Yes

Reviewer #2: Yes

5. Is the manuscript presented in an intelligible fashion and written in standard English?

Reviewer #1: Yes

Reviewer #2: Yes

6. Review Comments to the Author

Reviewer #1: I am satisfied with the current state of the paper and recommend it is accepted for publication.

Comment 1: I commend the authors for catching the mistake with reporting the effect size estimate.

Comment 2: In future studies it might be worthwhile to add a quick subjective measure of participants' self-perceived walking balance. That will help disambiguate meaningful disruptions to balance from mild or imperceptible disruptions.

Reviewer #2: • The authors conducted a secondary data analysis from a parent study to further examine whether the rope pulley system disrupts walking balance in healthy adults. They hypothesized that the rope pulley system may increase the variability of at least one of step width, step length, and step time which may indicate the disruption of walking balance. The result indicates increased variability in step time (M = 5 ms, p < 0.05, Cohen’s d = 1.286) but not in other parameters. The authors conclude that healthy subjects experienced minimal to moderate disruptions in walking balance when using with this device. This study addresses an interesting topic, and the manuscript is well written. However, given that the analysis is based on a small sample size and the significant result was found in only one of three variables of interest, there remains a couple of shortcomings in this study to be published under the current condition.

• The finding reported in this study is that the rope pulley system marginally increased step time variability but not all other parameters of interest. I think this is comparably weak findings with a small sample size. The variability parameters are all linear variability and non-linear gait dynamics will be a great addition to the current weak findings.

• Please provide the rationale for the hypothesis. How can an increase in either one of three parameters of variability be enough to indicate disruption in walking balance? Why not two or three but one?

• How were the variability parameters computed? Provide citation for the way of normalized standard deviation with leg length. Standard deviation also needs to be normalized with average value, which is a common computation of gait variability called coefficient of variation. Please consider using this parameter and justify the current data processing method.

• Standard deviation-based variability parameters are linear variability of human gait. Please consider additional analysis of non-linear analysis of gait dynamics (e.g., detrended fluctuation analysis), which is doable with 280 steps (10.1371/journal.pone.0174144) and thus a great addition to linear variability, so will strengthen the findings of this study.

7. PLOS authors have the option to publish the peer review history of their article (what does this mean?). If published, this will include your full peer review and any attached files.

Reviewer #1: No

Reviewer #2: No

---

## [Author Response · Author response to Decision Letter 1]

9 Feb 2022

Please note that we have included responses to the reviewer in the "Response to Reviewers" MS word document.

---

## [Decision Letter · Decision Letter 2]

8 Mar 2022

Step-to-step variability indicates disruption to balance control when linking the arms and legs during treadmill walking

PONE-D-21-31057R2

Dear Dr. Arellano,

We’re pleased to inform you that your manuscript has been judged scientifically suitable for publication and will be formally accepted for publication once it meets all outstanding technical requirements.

Kind regards,

David J Clark

Academic Editor

PLOS ONE

Additional Editor Comments (optional):

Reviewers' comments:

Reviewer's Responses to Questions

**Comments to the Author**

1. If the authors have adequately addressed your comments raised in a previous round of review and you feel that this manuscript is now acceptable for publication, you may indicate that here to bypass the “Comments to the Author” section, enter your conflict of interest statement in the “Confidential to Editor” section, and submit your "Accept" recommendation.

Reviewer #2: All comments have been addressed

2. Is the manuscript technically sound, and do the data support the conclusions?

Reviewer #2: Yes

3. Has the statistical analysis been performed appropriately and rigorously? 

Reviewer #2: Yes

4. Have the authors made all data underlying the findings in their manuscript fully available?

Reviewer #2: Yes

5. Is the manuscript presented in an intelligible fashion and written in standard English?

Reviewer #2: Yes

6. Review Comments to the Author

Reviewer #2: The authors have successfully addressed the concerns raised. The results reported substantiate that the rope pulley system may increase the step time variability which may indicate the disruption of walking balance.

7. PLOS authors have the option to publish the peer review history of their article (what does this mean?). If published, this will include your full peer review and any attached files.

Reviewer #2: No

---

## [Editor Report · Acceptance letter]

14 Mar 2022

PONE-D-21-31057R2 

Step-to-step variability indicates disruption to balance control when linking the arms and legs during treadmill walking 

Dear Dr. Arellano:

I'm pleased to inform you that your manuscript has been deemed suitable for publication in PLOS ONE. Congratulations! Your manuscript is now with our production department. 

Kind regards, 

on behalf of

Dr. David J Clark 

Academic Editor

PLOS ONE